# Deciphering the Role of Heme Oxygenase-1 (HO-1) Expressing Macrophages in Renal Ischemia-Reperfusion Injury

**DOI:** 10.3390/biomedicines9030306

**Published:** 2021-03-16

**Authors:** Maxime Rossi, Kéziah Korpak, Arnaud Doerfler, Karim Zouaoui Boudjeltia

**Affiliations:** 1Department of Urology, CHU de Charleroi, Université libre de Bruxelles (ULB), 6000 Charleroi, Belgium; arnaud.doerfler@chu-charleroi.be; 2Laboratory of Experimental Medicine (ULB 222 Unit), CHU de Charleroi, Hôpital André Vésale, Université libre de Bruxelles (ULB), 6110 Montigny-le-Tilleul, Belgium; keziah.korpak@chu-charleroi.be; 3Department of Geriatric Medicine, CHU de Charleroi, Hôpital André Vésale, Université libre de Bruxelles (ULB), 6110 Montigny-le-Tilleul, Belgium

**Keywords:** macrophage polarization, HO-1, renal IRI, AKI

## Abstract

Ischemia-reperfusion injury (IRI) is a leading cause of acute kidney injury (AKI), which contributes to the development of chronic kidney disease (CKD). Renal IRI combines major events, including a strong inflammatory immune response leading to extensive cell injuries, necrosis and late interstitial fibrosis. Macrophages act as key players in IRI-induced AKI by polarizing into proinflammatory M1 and anti-inflammatory M2 phenotypes. Compelling evidence exists that the stress-responsive enzyme, heme oxygenase-1 (HO-1), mediates protection against renal IRI and modulates macrophage polarization by enhancing a M2 subset. Hereafter, we review the dual effect of macrophages in the pathogenesis of IRI-induced AKI and discuss the critical role of HO-1 expressing macrophages.

## 1. Introduction

Acute kidney injury (AKI) is defined by the abrupt loss of renal function that is frequently associated with poor outcomes such as prolonged length of both intensive care and hospital stays, advanced chronic kidney disease (CKD), and even death [1]. The incidence of AKI in hospitalized patients is generally in the 2–7% range, with an incidence up to 10% in the intensive care unit population [1,2]. Therefore, the development of therapeutic or preventive strategies for AKI is an important public health concern [3,4]. The causes of AKI are numerous and can be divided into three categories [1]: prerenal (caused by decreased perfusion of the kidney), renal (with direct intrinsic kidney damage), and postrenal (caused by an obstruction of the urinary tract). However, renal ischemia-reperfusion injury (IRI) represents a leading cause of AKI [3]. IRI is a two-step pathological condition characterized by an initial restriction of blood supply to an organ followed by subsequent restoration of perfusion and re-oxygenation [5]. The kidney is one of the most susceptible organ to IRI [6]. Indeed, IRI is inherent to renal transplantation and leads to delayed graft function (DGF) of transplanted kidneys from deceased donors in up to 20 to 50% of cases [3,7]. The pathophysiology of IRI-induced AKI is very complex and combines major ischemia-induced cell stress, a significant burst of free radicals, and strong inflammatory immune responses leading to extensive cell injury, tissue damage, and subsequent kidney dysfunction [3,8,9]. In this context, macrophages play a critical role in IRI-induced AKI by exhibiting distinct phenotypes, which contribute to either inflammation, tissue injury or kidney repair [10]. Focusing on mice literature, this review summarizes the dual effect of macrophages on renal IRI and analyzes the role of the heme oxygenase-1 (HO-1) cytoprotective pathway as an emerging target for understanding the macrophage phenotypic switch. We further decipher HO-1 expressing macrophages acting as key players in IRI-induced AKI.

To better understand the impact of macrophages in renal IRI, it is essential to briefly discuss the role of renal tubular epithelial cells (RTECs) and other myeloid cells in the pathogenesis of IRI-induced AKI.

## 2. Tubular Cells and IRI-Induced AKI

RTECs are the cornerstone of the immune response in ischemic AKI [3]. During IRI-induced AKI, injured or dead RTECs release many endogenous molecules termed damage-associated molecular patterns (DAMPs) into the extracellular compartment [5]. These ligands (e.g., high-mobility group box 1 (HMGB1), heat shock proteins (HSPs), ATP) may bind to the Toll-like receptors (TLRs) expressed on RTECs, such as TLR2 and TLR4, and further induce the release of proinflammatory cytokines and chemokines (e.g., IL-1β, IL-6, tumor necrosis factor-α (TNF-α), monocyte chemoattractant protein-1 (MCP-1), and IL-8) through activation of TLRs downstream pathways (i.e., nuclear factor-κB (NF-κB), mitogen-activated protein kinase (MAPK) and type I interferon pathways) [5,6,7,8,9,10,11]. These chemokines and cytokines are crucial mediators for the recruitment and activation of innate immune cells into the postischemic kidney [11]. Interestingly, TLR2 and TLR4 expression is increased upon renal IRI that may amplify the inflammatory response [12]. During ischemic AKI, the damaged RTECs release huge amounts of reactive oxygen species (ROS), which result in oxidative stress leading to impairment of mitochondrial oxidative phosphorylation and subsequent adenosine triphosphate (ATP) depletion [13]. Oxidative stress plays a critical role in the pathogenesis of IRI-induced AKI [14]. Indeed, oxidative stress increases the expression of NO and superoxide, which both rapidly react to generate peroxynitrite anion, a nitrating and oxidizing agent, resulting in oxidative damage to proteins, lipids, carbohydrates, and DNA [14]. Adaptive immunity may be implicated in IRI-induced AKI through the tubular epithelium. Indeed, the proximal tubular epithelial cells express major histocompatibility complex class II molecules (MHC II) and costimulatory molecules (i.e., B7-1 and B7-2) and may therefore present antigen to T lymphocytes [3,4,5,6,7,8,9,10,11,12,13,14,15,16].

## 3. Myeloid Cells and IRI-Induced AKI

Myeloid cells derive from hematopoietic stem cells in the bone marrow (BM) and include granulocytes and monocytes [17]. Neutrophils represent the most abundant type of granulocytes and the others (i.e., eosinophils and basophils) will not be discussed hereafter. Circulating monocytes differentiate into tissues macrophages with location-dependent specific functions (e.g., the Kupffer cells in the liver, mesangial macrophages in the kidneys, and alveolar macrophages in the lung) or into dendritic cells (DCs) in lymphoid organs [17]. Upon danger signals or pathogen invasion, myeloid cells can be rapidly activated and recruited to injured tissues where they release inflammatory cytokines [17]. Then, macrophages and DCs may also present antigens to effector T cells and trigger alloreactivity. Myeloid cells have, therefore, a critical role in both innate and adaptive immune responses [18]. Interestingly, myeloid cells can be important contributors to the pathogenesis of IRI-induced AKI [19].

Neutrophils are the most abundant circulating white blood cells. They represent key effector cells of the innate immune system that modulate the earliest inflammatory responses to pathogens through release of cytotoxic proteases and ROS. Massive influx of neutrophils has been described in postischemic kidney and thought to be the onset of tubular injury [20,21]. Indeed, they begin to infiltrate the kidney about 30 min after reperfusion, particularly in the outer medulla [3]. Damaged endothelial cells express a huge amount of cell adhesion molecules (e.g., ICAM-1, E-selectin, L-selectin, and integrins) leading to increased endothelium–leukocyte interactions [22]. Subsequently, this neutrophil–endothelium interaction induces capillary occlusion and vascular congestion of the renal microcirculation, which amplifies oxygen deprivation and renal tissue destruction [12,13,14,15,16,17,18,19,20,21]. Furthermore, neutrophils may also transmigrate into the interstitium. Surrounding renal tubules, neutrophils release proteases, ROS, and cytokines (e.g., IL-1, IL-6, IL-17, TNF-α) that increase endothelial dysfunction, and impair both epithelial and endothelial architecture with magnification of renal tissue injury [3,4,5,6,7,8,9,10,11,12,13,14,15,16,17,18,19,20,21]. Neutrophils may also positively regulate their transmigration through a positive feedback loop between IL-17 and interferon (IFN)-γ [23]. Then, inhibiting neutrophil infiltration into postischemic kidney has been shown to mitigate IRI [22,23,24]. Finally, neutrophils are involved in the pathogenesis of IRI-induced AKI by obstructing renal microcirculation and releasing ROS, proteases, and cytokines.

Renal DCs arise from common progenitor cells in the BM [25]. The renal CD11c^+^ MCH II^+^ DC population is complex and expresses various levels of CD11b and F4/80 [26]. CD11c^+^ MCH II^+^ DCs can be separated into two distinct subsets: CD103^+^ cells (i.e., CD103^+^ CD11b^lo^ CD135^+^ CX3CR1^−^ F4/80^−^) and CD11b^+^ cells (i.e., CD103^−^ CD11b^+^ CD115^+^ CX3CR1^+^ F4/80^+^) [25,27]. The origin of monocytes and their differentiation to macrophages and DCs will be discussed below in the following section. During IRI-induced AKI, kidney-resident DCs acts as sentinel by detecting DAMPs. Then, these cells produce proinflammatory cytokines and chemokines such as TNF-α, suggesting a proinflammatory role for DCs in renal IRI [28]. However, some studies have shown that renal DCs mitigated renal tissue damage, suggesting an anti-inflammatory effect [29,30]. After sensing DAMPs, matured DCs induce adaptive immunity. These cells migrate to draining lymph nodes for presenting antigens to specific T cells, which are released into circulation to infiltrate injured kidney [31].

## 4. Macrophages and IRI-Induced AKI

### 4.1. Origins of the Monocytes/Macrophages

Macrophages and DCs arise from common progenitor cells in the BM under the control of key growth factors: colony-stimulating factor 1 (CSF-1, also known as macrophage colony-stimulating-factor, M-CSF), fms-like tyrosine kinase 3 ligand (Flt-3L), granulocyte macrophage colony-stimulating factor (GM-CSF) [25]. The main growth factor axes are Flt-3L/CD135 (also known as fms-like tyrosine kinase 3 receptor, Flt-3) and CSF-1/CD115 (also known as colony-stimulating factor 1 receptor, CSF-1R) [25]. Two types of CD11b^+^ CD115^+^ monocyte subsets have been identified in mice [32]. “Classical” monocytes (also termed inflammatory monocytes) are defined by the surface marker combination CD11b^+^ CCR2^hi^ GR-1^int^ Ly6C^hi^ CX3CR1^int^ CD43^lo^ CD62L^+^ [32,33,34]. These inflammatory monocytes are recruited to inflamed tissues, such as injured kidney, or infection site and differentiate into macrophages and DCs [32,33,34]. However, this Ly6C^hi^ BM-derived monocyte subset may also contribute to the resident macrophages and DCs pool at steady state [25]. In contrast, “non-classical” monocytes (also termed patrolling monocytes) are characterized by the surface marker combination CD11b^+^ CCR2^lo^ GR-1^−^ Ly6C^lo^ CX3CR1^hi^ CD43^+^ CD62L^−^ [32,33,34]. Due to high expression of adhesion-related receptor CX3CR1, this monocyte subset exhibits the ability to patrol in the bloodstream and migrates to healthy tissues where they differentiate into resident macrophages and DCs [32,33,34]. These patrolling monocytes also contribute to the endothelial cell homeostasis by scavenging luminal microparticles and debris [33,34]. “Classical and non-classical” subsets are represented equally in mice [33]. The subsequent macrophages and DCs represent the renal mononuclear phagocytes (rMoPh) that play a critical role in the kidney [25].

### 4.2. Involvement of Distinct Macrophages in Renal IRI

During renal IRI, resident rMoPh may release proinflammatory cytokines (e.g., TNF-α, IL-1, IL-6) and chemokines (e.g., CCL2, CCL5, CXCL10, CXCL2) [25,28,35,36]. Therefore, Ly6C^+^ monocytes infiltrate the injured kidney through a CCL2/CCR2 signaling pathway with a small proportion of circulating Ly6C^−^ monocytes [25,35,37]. One hour after reperfusion, the influx of macrophages is increased in the injured kidney with a peak at 24 h and remains for 7 days [35]. Macrophages accumulate in the outer medulla of the postischemic kidney [10].

Distinct subsets of macrophages may occur in kidney and tissue macrophages derived from infiltrating monocytes can undergo a switch to different phenotypes depending on microenvironment [38] (Figure 1). In response to DAMPs/proinflammatory mediators, infiltrating Ly6C^+^ monocytes may differentiate into classically activated macrophages (i.e., M1 macrophages), which express proinflammatory phenotype [3,10,25]. M1 macrophages are induced by exposure to lipopolysaccharide (LPS), IFN-γ, TNF-α, or GM-CSF [10,38,39]. These inflammatory mediators are released in renal interstitium by neighboring immune cells (i.e., neutrophils, NK cells, Th1/Th17 cells) [10]. Then, M1 macrophages release proinflammatory cytokines (e.g., TNF-α, IL-1β, IL-6), ROS that further amplify IRI-induced AKI through a positive feedback loop [10,38,40]. Indeed, M1 macrophages contribute also to the recruitment of neutrophils, and induction of epithelial cells apoptosis [10]. These M1 macrophages can be identified by their high expression of inducible nitric oxide synthase 2 (iNOS), IL-12, IL-23, and Ly6C [38] (Figure 1). M1 macrophages display a proinflammatory phenotype with strong antimicrobial activity and promote or amplify Th1 polarization of CD4^+^ T cells by IL-12 release [41]. Interestingly, depletion of kidney macrophages by liposomal clodronate (LC) at the early stages of IRI reduces AKI and improves renal repair, suggesting a critical role for macrophages in IRI-induced AKI [29,30,31,32,33,34,35,36,37,38]. Moreover, adoptive transfer of IFN-γ-stimulated macrophages in LC-treated IRI mice worsen AKI, suggesting the pathogenic role of M1 macrophages in ischemic AKI [38].

Subsequently to the early phases of IRI, Th2 and regulatory T (Tregs) cells are recruited in the injured renal tissue and produce high levels of IL-4, IL-10 and IL-13 [10,11,12,13,14,15,16,17,18,19,20,21,22,23,24,25,26,27,28,29,30,31,32,33,34,35,36,37,38] (Figure 1). This exposure to Th2-type cytokines (i.e., IL-4 and IL-13) results in a macrophage switch to anti-inflammatory M2 phenotype (also termed alternatively activated macrophages) characterized by high expression of arginase-1 (Arg1), mannose receptor (MR, also termed CD206), chitinase-like protein (e.g., Ym1), resistin-like protein (Fizz1), CD36 (fatty acid translocase), and IL-10 associated with down-regulated expression of proinflammatory markers (i.e., IL-12 and iNOS) [10,11,12,13,14,15,16,17,18,19,20,21,22,23,24,25,26,27,28,29,30,31,32,33,34,35,36,37,38,39,40,41] (Figure 1). Notably, M2 macrophages can occur through a switch from M1 to M2 phenotype or directly from infiltrating monocytes [38]. In addition, macrophage uptake of apoptotic cells releasing high levels of anti-inflammatory cytokines (i.e., TGF-β and IL-10), associated with reduction in DAMPs, produce a tissue microenvironment that would promote macrophage polarization towards the M2 profile [42,43,44]. M2 macrophages display an anti-inflammatory profile and play a critical role in anti-parasite immune response, wound healing, and fibrosis [45,46]. M2 macrophages may further be subdivided into three different subsets: M2a (induced by exposure to IL-4 or IL-13), M2b (induced by stimulation with immune complexes such as LPS or IL-1β), and M2c (induced by IL-10, TGF-β, or glucocorticoids). M2a and M2b macrophages promote a Th2 immune response while M2c macrophages are involved in tissue remodeling and display regulatory properties [10,41]. Although they have been described in vitro, these different subtypes of macrophages (i.e., M1, M2a, and M2b subsets) do not reflect their real function in vivo [10]. Macrophages seem to display different phenotypes in response to microenvironment rather than be separated into stable subpopulations. In line with that, a recent study identifies unique macrophage populations according to differential Ly6C expression [47]. In this study, the CD11b^+^ Ly6C^hi^ subset is associated with early stages of renal injury and subsequent proinflammatory state, whereas the CD11b^+^ Ly6C^int^ subset predominates during proliferative repair phase. The CD11b^+^ Ly6C^lo^ subset emerges with renal fibrosis. The authors also show that the Ly6C^int^ and Ly6C^lo^ subpopulations do not fit into the M1/M2 classification as defined in vitro. Finally, these three different subsets are identified by unique gene signature that provides insight into their function in the pathophysiology of IRI-induced AKI [47]. This concept probably reflects more the in vivo situation than the M1/M2 paradigm.

The mechanisms enabling macrophage change from the M1 to M2 subset remain unclear. Interestingly, macrophage phenotypic switch to M2 can be also induced by either RTECs or apoptotic cell-derived factors, such as CSF-1 and sphingosine-1-phosphate (S1P), respectively [10,48,49].

### 4.3. Macrophages and Renal Repair after IRI

AKI is considered as a reversible process with subsequent complete recovery of the kidney [50]. When renal insult is slight, the repair mechanism may be adaptive with few long-term impairments. After IRI-induced AKI, RTECs lose their polarity and brush border, mainly in the proximal tubule, leading to tubule cell death [50]. During adaptive repair, surviving RTECs undergo dedifferentiation and proliferation to restore the integrity and functionality of nephron [51,52]. Moreover, pericytes remain in close proximity to the capillary system and reduce myofibroblast proliferation, which hence minimizes resultant renal fibrosis [4].

Macrophages play an important role in adaptive repair by phagocyting both dying RTECs and neutrophils [4,53]. The early influx of M1 macrophages promote a proinflammatory state useful to remove damaged or died RTECs and neutrophils [38]. At 3 to 5 days after injury, a phenotypic switching of macrophages occurs with an Arg1^+^ M2 predominance, which play a critical role during the repair phase of renal IRI [38]. Furthermore, depletion of macrophages, by either LC or diphtheria toxin (DT, used in transgenic mice expressing the human diphtheria toxin receptor, DTR, under the control of CD11b promoter), during this recovery phase is associated with persistent renal inflammation, decreased RTECs proliferation, and delayed tubule repair [30,38,54]. M2 macrophage-derived regenerative molecules are not well known. However, macrophage-derived wingless-related MMTV integration site 7B (Wnt7b) has been shown to promote kidney regeneration through epithelial cell-cycle progression and tubule basement membrane repair [55]. Thus, M2 macrophages also release chitinase-like protein breast regression protein-39 (BRP-39) (Figure 1). This macrophage-derived mediator acts on RTECs to activate the phosphatidylinositol 3-kinase/protein kinase B (PI3K/Akt) pathway and inhibit ROS-mediated tubular apoptosis [56].

### 4.4. Macrophages and Fibrogenesis after IRI

Despite ability to recover after AKI, injured kidney is often associated with maladaptive repair leading to impairment in renal structure and function [4]. Several risk factors have been identified for the progression of AKI to CKD, such as the severity, duration and frequency of AKI episodes, age, preexisting CKD, and other comorbidities (e.g., diabetes) [57]. During maladaptive repair, renal inflammation remains uninterrupted leading to pericyte dissociation from capillaries and subsequent fibroblast proliferation-induced deposition of collagen [4,50].

Macrophages may also contribute to kidney fibrosis upon IRI-induced AKI [10]. Indeed, in a unilateral ureteral obstruction (UUO) model, macrophage depletion using LC mitigates RTECs apoptosis and subsequent renal fibrosis [58]. Additionally, renal interstitial fibrosis is reduced upon UUO in DT-treated CD11b-DTR mice while renal scarring is not attenuated in DT-treated CD11c-DTR mice [59,60]. M1 macrophages release proinflammatory molecules, such as TNF-α and ROS, which induce renal inflammation and subsequent tissue fibrosis [10]. Moreover, M1 macrophages promote kidney fibrosis by secretion of matrix metalloproteinase-9 (MMP-9), which modulates tubular cell epithelial–mesenchymal transition (EMT) [61,62] (Figure 1).

M2 macrophages mitigate renal inflammation by secretion of anti-inflammatory cytokines, such as IL-10 and TGF-β [10]. Interestingly, TGF-β is a major cytokine driving the differentiation of quiescent fibroblasts into active myofibroblasts, suggesting a key role of M2 macrophages in fibrogenesis [10,63] (Figure 1). Furthermore, M2 macrophages also release galectin-3, which induces renal fibrosis [64] (Figure 1). Recently, several studies have pointed out that macrophages may directly differentiate into collagen-producing myofibroblasts in both human and mouse renal fibrosis, suggesting that macrophage-to-myofibroblast transition (MMT) may be a direct pathway leading to fibrogenesis [65,66,67,68]. Noteworthy, MMT cells have a predominant M2 phenotype [67,68]. Altogether, M1 and M2 macrophages may promote renal fibrosis through direct and indirect pathways.

## 5. Heme Oxygenase-1 (HO-1)

### 5.1. Overview

In 1968, Tenhunen and Schmid described the catalytic breakdown of heme by a microsomal enzyme called heme oxygenase (HO) [69,70]. HO, a heme-containing HSP (also named HSP32), metabolizes free heme into carbon monoxide (CO), iron, and biliverdin, which is converted to bilirubin by biliverdin reductase [71]. To date, three isoforms of HO have been identified: HO-1, HO-2, and HO-3 [72]. The last isoform, HO-3, was discovered in the rat brain [73]. HO-3 is thought to be catalytically inactive and be involved in heme sensing or binding. Its properties remain unclear [73]. HO-2 is a constitutive isoform and mainly found in the brain, testes and endothelium [74]. HO-2 is expressed under homeostatic condition and contributes to protection against oxidative stress [74]. HO-1 is the inducible isoform activated within hours of exposure to cellular stress inducers, such as pathogens, oxidants, hypoxia, inflammatory chemokines/cytokines, tissue damage [71,74]. Both HO-1 and HO-2 catabolize heme degradation [75].

HO-1 is located in the endoplasmic reticulum, the inner mitochondrial membrane, and plasma membrane caveolae [71]. The enzyme is encoded by the *Hmox1* gene, which shares similar architecture in human, mouse, and rat [71].

### 5.2. Regulation of HO-1 Expression

The nuclear factor erythroid 2-related factor 2 (Nrf2, activator) and the BTB and CNC homology 1 (Bach1, repressor) are redox-dependent transcription factors, which play a central role for HO-1 induction in response to oxidative stress [76]. Under basal conditions (e.g., low level of intracellular heme), Bach1 binds to stress-responsive element motifs of the *Hmox1* promoter and represses the expression of HO-1 [77]. Heme regulates the cellular level of Bach1 through a proteasomal degradation [78]. Moreover, Nrf2 expression is naturally repressed by the cytosolic inhibitor Kelch-like ECH-associated protein 1 (Keap1) [79,80]. In case of high intracellular level of heme and/or stress stimuli exposure, Nrf2 dissociates from Keap1, which allows its subsequent nuclear translocation [81,82]. In the nucleus, Bach1 is removed from the Hmox1 promoter, which enables Nrf2 to bind to stress-responsive element motifs and thus to induce HO-1 expression [77].

### 5.3. Cytoprotective Effects of HO-1

In 1994, R. Tyrrell and colleagues described for the first time that induction of HO-1 generated an adaptive cytoprotective response to oxidative stress in cultured human fibroblasts [83].

Initially, the impact of HO-1 in oxidative stress was identified in cultured *Hmox1*-deficient embryonic fibroblasts, which exhibited higher production of free radicals in response to prooxidant agent exposure (i.e., hemin or hydrogen peroxide) as compared to wild-type embryonic fibroblasts [84]. The antioxidant effects of HO-1 are thought to come from the catabolism of free heme [85,86]. Indeed, free heme is mainly produced through oxidation of hemoproteins (e.g., hemoglobin and myoglobin) [85,87]. Then, free heme may act as a Fenton reactor to produce toxic hydroxyl radicals released from hydrogen peroxide [88]. These ROS may damage DNA and proteins, which lead to programmed cell death by apoptosis [89]. Therefore, the degradation of free heme through HO-1 limits the production of subsequent prooxidant and cytotoxic agents [90].

Several studies have shown that HO-1 expression protects different types of cells from apoptosis [91,92,93]. The antiapoptotic effect of HO-1 is mainly associated with the generation of CO through a p38 MAPK-dependent pathway [94]. Indeed, HO-1 expression induces the degradation of the p38α MAPK apoptotic isoform by the proteasome pathway with sparing of the p38β MAPK antiapoptotic isoform [95,96,97]. Furthermore, activation of the p38 MAPK pathway by HO-1 also modulates expression of the antiapoptotic molecule B-cell lymphoma-extra large (Bcl-xL), which may inhibit the intrinsic (mitochondrial) apoptotic pathway [98,99].

### 5.4. Anti-Inflammatory Effect of HO-1

In 1996, D.A. Willoughby and colleagues identified for the first time that HO-1 may modulate the immune response [100]. Indeed, in a rat model of pleurisy, they showed that HO-1 upregulation mitigated the inflammation (i.e., a reduced leukocyte influx in pleural cavity), whereas its downregulation led to an exacerbated immune response [100]. Interestingly, compelling evidence suggests a positive feedback loop between HO-1 and IL-10, the well-known anti-inflammatory cytokine, especially in monocytes/macrophages [101,102,103]. Through its receptor, IL-10 phosphorylates signal transducer and activator of transcription 3 (STAT3), which translocates to the nucleus, resulting in HO-1 induction [101,104]. Then, HO-1 mediates the anti-inflammatory effect of IL-10, as suggested by an attenuation of IL-10-induced protection in a mice LPS septic shock model with concomitant inhibition of HO-1 expression [102]. HO-1 and its byproduct CO may also modulate IL-10 production through the activation of p38 MAPK pathway, therefore suggesting an IL-10/HO-1 axis [101,105]. On macrophages, CO has an anti-inflammatory effect via inhibition of TLRs signaling pathways in response to LPS [106].

## 6. HO-1 Expressing Macrophages and Renal IRI

Several natural cellular mechanisms may confer resistance against renal IRI, including the HO-1 cytoprotective pathway [71]. Interestingly, HO-1-deficient mice exhibit severe AKI and death upon renal IRI [19,107]. Conversely, prior HO-1 induction with synthetic heme (i.e., hemin) may confer significant resistance against renal IRI [9,108].

### 6.1. Macrophages Are Critical for HO-1 Cytoprotective Effects

Until recently, both epithelial (i.e., tubular cells) and endothelial cells were commonly believed to represent the critical source of HO-1 during IRI-induced AKI. This hypothesis was mainly supported by the intense susceptibility of fully HO-1-deficient to renal IRI [107,109,110]. However, Ferenbach DA et al. already demonstrated that genetically modified or hemin-induced HO-1^+^ macrophages mediate protection against renal IRI [111,112]. Hull et al. showed that HO-1 is a critical regulator of the trafficking of myeloid cells in AKI [19]. In addition to these previous studies, it has been shown that, in response to IRI, naturally occurring HO-1 expressing macrophages may already modulate the severity of AKI [108]. Indeed, even if the global expression of HO-1 in the whole kidney is not affected, the absence of HO-1 expressing macrophages is critical in the outcome of renal IRI [108]. Thus, HO-1 expressing macrophages is identified as a critical regulator of the earliest phases of IRI (i.e., lower plasma creatinine, tubular damage, and renal inflammation) that may mitigate the risk of severe AKI upon IRI [108]. Moreover, hemin-mediated protection requires specific expression of HO-1 within myeloid cells. CD11b^+^ F4/80^lo^ macrophages are identified as the main protective myeloid source of HO-1 upon renal IRI. Indeed, hemin preconditioning specifically upregulates HO-1 within these myeloid cells [108].

### 6.2. HO-1 and Macrophage Polarization

HO-1 expression is associated with CD11b^+^ F4/80^lo^ macrophages that exhibit regulatory properties (i.e., “M2” macrophages) [108,113]. In 2009, N. Weis and colleagues first described the involvement of HO-1 in macrophage polarization toward an M2 phenotype [114]. Moreover, a study investigating the role of Bach1 (repressor of HO-1 expression) in inflammatory bowel disease has identified that macrophages from Bach1-deficient mice exhibit an M2 profile (i.e., expression of M2 markers such as Arg1, Ym1, and Fizz1) with concomitant HO-1 overexpression [115]. Then, a recent study has shown that myeloid HO-1 modulates macrophage polarization and protects against liver IRI by enhancing a M2 anti-inflammatory phenotype [116]. M2 macrophages modulate inflammatory responses upon renal IRI and promote tissue repair after insult [38]. In this condition, HO-1 may foster a microenvironment in favor of M2 phenotype that efficiently mitigates AKI and prevents transition to CKD [101].

Interestingly, HO-1 controls IL-10 expression and HO-1^+^ macrophages release a high level of IL-10, suggesting a close relation between these two mediators, promoting macrophage phenotype switch to M2 [102,105,111].

The intense inflammatory response observed in the absence of HO-1 expressing macrophages upon renal IRI may be explained through a phenotypic polarization toward “M1” macrophages [108]. Indeed, HO-1 inhibition/deletion is associated with a lack of M2 macrophages and a simultaneous excess of M1 inflammatory macrophages [117]. Then, these macrophages secrete pro-inflammatory mediators that amplify intrarenal inflammation and injury through interaction with kidney resident cells [10,38]. 

Thus, HO-1 influences a switch to M2 phenotype, which may explain, at least in part, its anti-inflammatory properties. However, the precise molecular mechanism of macrophage polarization mediated by HO-1 remains unclear and requires further investigation [101].

### 6.3. HO-1 Expressing Macrophages Mitigates Distant Organ Injuries upon Renal IRI

Renal IRI releases pro-inflammatory cytokines (e.g., IL-1β, IL-6, and TNF-α) into systemic circulation leading to inflammatory cell recruitment and remote organ injuries [118]. Acute lung injury (ALI) is the most frequent distant insult related to AKI and the mortality significantly rises when both diseases coexist [119,120]. IRI-induced AKI promotes the occurrence of ALI [118]. Subsequent systemic inflammation contributes to affect alveolar and pulmonary interstitial spaces. Endothelium is therefore activated with disruption of vascular barrier. This imbalance results in leukocytes transmigration into pulmonary interstitium [118]. The inflammatory infiltrate aggravates ALI through pro-inflammatory storm, oxidative damage, and apoptosis [118,119]. Interestingly, an antimalarial drug (i.e., artesunate) prevents AKI-induced ALI through HO-1 expression [121]. In line with this result, hemin-induced HO-1^+^ macrophages dampen systemic inflammatory responses and mitigate AKI-induced ALI by limiting lung inflammation [9].

### 6.4. HO-1 Expressing Macrophages Modulates Adaptive Renal Repair after AKI

A cell-cycle arrest at the G2/M phase is associated with maladaptive repair and subsequent fibrosis in renal IRI [4,122]. The roles of cell-cycle inhibitors p53/p21 in the pathogenesis of AKI remain unclear. Indeed, p53 release by leukocytes protects kidney against AKI, whereas its expression in RTECs is associated with severe AKI and higher risk of CKD [123,124,125]. Then, p21 is known to promote cell-cycle arrest in the G1 phase, repair DNA-damage, and thus protects against renal IRI [126,127]. However, p21 fails to mitigate interstitial fibrosis upon AKI [127]. Otherwise, p21 is also a marker of RTECs cellular senescence reflecting lower regenerative ability and increased risk of kidney fibrosis following AKI [128,129]. 

Interestingly, absence of HO-1 expressing macrophages is associated with impaired renal repair upon IRI as suggested by the upregulation of cell-cycle regulatory proteins (i.e., p53, p21), and early interstitial fibrosis, a central marker of CKD [108].

Furthermore, HO-1^+^ macrophages-deficient mice also exhibit p62 accumulation upon renal IRI which may be seen as a surrogate marker of impaired autophagy, a phenomenon known to enhance interstitial fibrosis upon tubular stress [130]. Furthermore, these data suggest a link between HO-1^+^ macrophages deficiency and renal fibrosis because of maladaptive repair.

### 6.5. The Origin of HO-1 Expressing Macrophages

Both resident and infiltrating HO-1^+^ macrophages may protect kidney against IRI-induced AKI. Consistent with previous study [112], hemin induces HO-1 expression within renal CD11b^+^ F4/80^lo^ macrophages, even in normal conditions (i.e., absence of IRI) [108]. This result suggests an involvement of tissue-resident macrophages in the earliest phase of renal IRI. Interestingly, after renal IRI, hemin and saline-treated mice express same amount of HO-1 in the kidney, suggesting that, despite being a minor cellular source of HO-1, HO-1^+^ macrophages mediate significant renoprotection upon IRI [108]. 

It is well-known that splenic macrophages protect against AKI [131,132]. Interestingly, hemin induces HO-1 within spleen CD11b^+^ F4/80^lo^ macrophages, suggesting that extra-renal HO-1^+^ macrophages may constitute a pool that can be recruited in the ischemic kidney [108]. Indeed, one day after reperfusion, a higher amount of CD11b^+^ F4/80^lo^ macrophages is noted in the kidney of hemin-treated mice, suggesting that renoprotection may also be provided by recruited HO-1^+^ macrophages [108]. 

In term of remote organ injuries following renal IRI, HO-1 mitigates AKI-induced ALI. Resident alveolar macrophages (AMs) may modulate inflammation and promote tissue healing through multiple anti-inflammatory pathways including HO-1 [133]. Furthermore, primary rat AMs express high levels of HO-1 after in vitro hemin exposure [133]. Accordingly, improving outcomes of AKI-induced ALI by hemin may be explained through HO-1 expressing lung-resident macrophages [9]. As observed in the kidney, there is no HO-1 upregulation in the whole lung, suggesting that HO-1^+^ macrophages represent a functionally important cell population [9]. Otherwise, worsened systemic inflammation and ALI are observed in splenectomized mice after renal IRI due to decreased splenic IL-10 production [134]. Interestingly, HO-1^+^ macrophages release huge amounts of IL-10 [111]. Therefore, HO-1^+^ spleen macrophages may mitigate systemic inflammatory response and constitute a reservoir with a potential recruitment to ischemic kidney and distant injured organs for limiting insults. Accordingly, we postulate that both tissue-resident and infiltrating/circulating HO-1^+^ macrophages modulate HO-1-mediated improvement after IRI-induced AKI.

## 7. Macrophages and IRI-Induced AKI in Humans

In humans, three monocyte subsets have been described according to differential expression of CD14 and CD16 on HLA-DR^+^ cells: CD14^+^ CD16^−^ “classical” monocytes (80–90% of the human monocyte pool), CD14^+^ CD16^+^ intermediate and CD14^lo^ CD16^+^ “non-classical” monocytes [33,34]. These human monocytes exhibit same properties as described in mice. The intermediate subset generally expresses an inflammatory phenotype [34]. In human macrophages, CD68 is a general marker, whereas HLA-DR and CD163 are M1 and M2 markers, respectively [135].

Human macrophages present in normal or ischemic kidneys have been poorly investigated compared to those in mice, and, therefore, translation of animal findings to human disease remains difficult [136]. In biopsy specimens of human AKI, macrophages have been identified as the main cell type infiltrating the kidney and persist during tissue repair [38,137]. These macrophages surround injured RTECs and exhibit a M2 phenotype (i.e., CD163^+^ macrophages) [135]. However, macrophages also infiltrate renal allografts with proven acute cellular or chronic rejection and seem to be associated with poor outcomes [136,137,138,139]. Whether these data demonstrate a macrophage phenotype associated with inflammation or tissue repair remain to date unresolved and require further investigations.

## 8. HO-1 Expressing Macrophages: A Novel Nephron Sparing Strategy?

The synthetic heme protein, hemin, upregulates the HO-1 expression within tissue-resident and infiltrating/circulating macrophages with subsequent regulatory properties [108]. Interestingly, the pharmacological induction of HO-1 with hemin is effective in humans and well tolerated with a low rate of adverse events [140]. Furthermore, hemin has been used safely in humans for decades in the treatment of acute intermittent porphyria and recently in renal transplant [141,142].

Hence, hemin may be a harmless, novel, and promising approach to induce HO-1-expressing macrophages for limiting kidney damage with subsequent CKD, and distant organ injury after renal IRI.

## 9. Conclusions

In summary, macrophages display two divergent faces in the setting of IRI-induced AKI [38]. The early influx of macrophages promotes a proinflammatory state that amplifies tissue injuries. Then, in response to local signals, macrophages (i.e., M1, recruited monocytes or tissue-resident macrophages) undergo phenotypic switch to M2 macrophages that suppress renal inflammatory response and promote tissue repair. Indeed, depletion of macrophages before IRI mitigates renal insult, whereas depletion of macrophages 3 days after IRI delays renal tissue remodeling [38]. 

The anti-inflammatory enzyme, HO-1, influences the macrophage phenotypic switch towards a M2 subtype and confers resistance to IRI-induced AKI through specific expression within CD11b^+^ F4/80^lo^ macrophages. This myeloid cell sub-population is observed in the kidney and spleen, suggesting that protective effects may be provided by both tissue-resident and infiltrating/circulating HO-1^+^ macrophages. Moreover HO-1 expressing macrophages prevent maladaptive repair and subsequent CKD after renal IRI through modulation of cell-cycle and autophagy regulatory proteins. 

Then, HO-1 expressing macrophages play a critical role in the modulation of IRI-induced AKI by improving short- and long-term functional outcomes after renal IRI (summarized in Figure 2). Accordingly, modulation of HO-1 expressing macrophages may be an efficient preventive strategy for limiting kidney damage after renal IRI.

## Figures and Tables

**Figure 1 biomedicines-09-00306-f001:**
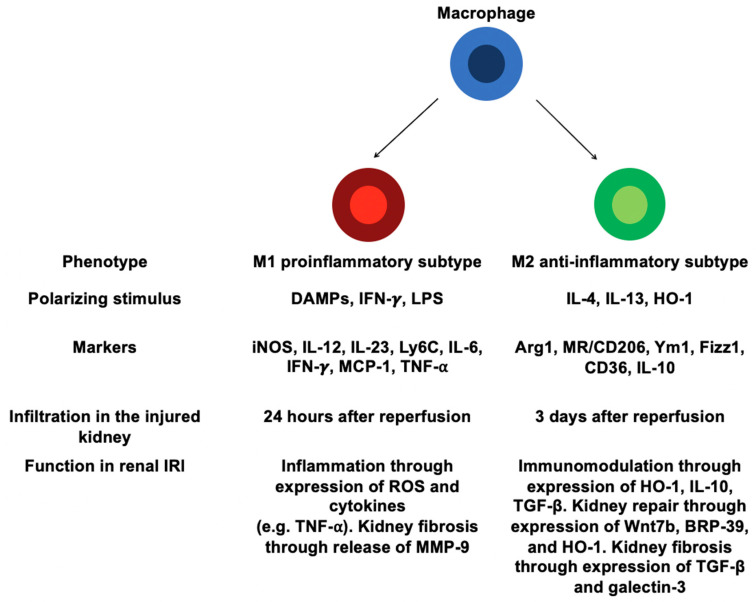
Macrophages in ischemia-reperfusion injury (IRI)-induced acute kidney injury (AKI). Distinct macrophage phenotypes are involved in renal injury and repair. Proinflammatory macrophages (M1) infiltrate the kidney 24 h after reperfusion and contribute to kidney injury. Anti-inflammatory macrophages (M2) are detected in the kidney 3 days after reperfusion. M2 macrophages dampen renal inflammation and promote tissue repair. Differentiation of tissue-resident macrophages or recruited monocytes into distinct macrophage subsets in response to local microenvironment. M1 macrophages contribute to inflammation by secretion of cytokines and reactive oxygen species (ROS). M1 macrophages may also promote kidney fibrosis through the release of MMP-9. M2 macrophages mediate kidney repair by secretion of Wnt7b, BRP-39, and heme oxygenase-1 (HO-1). Additionally, galectin-3 and TGF-β released by M2 macrophages induced renal fibrosis.

**Figure 2 biomedicines-09-00306-f002:**
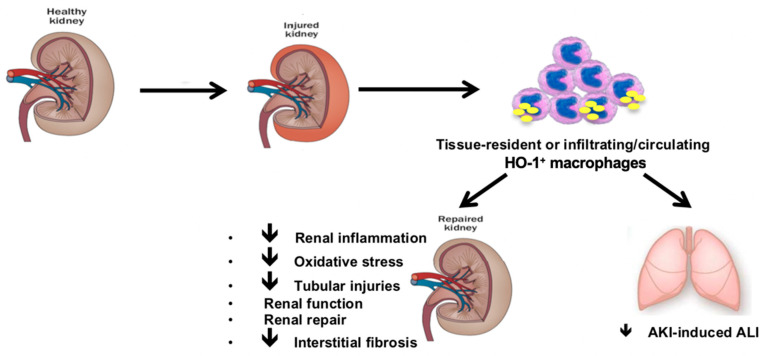
Role of HO-1 expressing macrophages in IRI-induced AKI. HO-1 expressing macrophages control the magnitude of renal IRI (i.e., less renal damage, renal inflammation and oxidative stress). Moreover, HO-1^+^ macrophages prevent maladaptive repair and subsequent chronic kidney disease (CKD) after renal IRI through modulation of cell-cycle and autophagy regulatory proteins. These anti-inflammatory macrophages also mitigate distant organ injury following renal IRI (e.g., AKI-induced acute lung injury (ALI)) by limiting systemic inflammatory response and remote organ inflammation. HO-1 expressing macrophages play, therefore, a critical role in the modulation of IRI-induced AKI by improving short- and long-term functional outcomes after renal IRI.

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
