# Peer review of "Deciphering the Role of Heme Oxygenase-1 (HO-1) Expressing Macrophages in Renal Ischemia-Reperfusion Injury"

_biomedicines, 2021, doi:10.3390/biomedicines9030306_

Round 1

Reviewer 1 Report

The review by Rossi et al is generally well-done and describes nicely macrophages involvement and polarization during AKI and how HO-1 is correlated with it.

I believe it will be useful for the field and I generally support its publication after addressing some small comments:

- What should be made clearer in the introduction is that this review mostly focuses on mice studies. To this end, It would increase the value of this review to have a paragraph before the conclusion that describes the knowledge now available in humans about macrophages involvement in AKI also describing the possible new or future strategy that aims to target HO-1.

- The description of monocyte subpopulations and the differentiation in Tissue-resident macrophages or inflammatory macrophages is based on old publications (i.e. Geissman 2003 ref 28). Even if those are the building block publications, would be appropriate to describe it in view also of newer publications and knowledge. (i.e: Development of monocytes, macrophages, and dendritic cells. Geissmann F, Manz MG, Jung S, Sieweke MH, Merad M, Ley K Science. 2010 Feb 5; 327(5966):656-61.; Guilliams et al Immunity. 2018 Oct 16;49(4):595-613; Nonclassical patrolling monocyte function in the vasculature.Thomas G, Tacke R, Hedrick CC, Hanna RN Arterioscler Thromb Vasc Biol. 2015 Jun; 35(6):1306-16.). Moreover, the resident macrophages embryonal origin or derived from circulating monocytes should be better described.

- Line 111 to 116 is exactly repeated in the subsequent paragraph line 132 to 138. Please check and rephrase accordingly.

- Line 142 the abbreviation rMoPh is introduced please describe it and be consistent in all the text.  

- Just as a stile note; starting a phrase of a paragraph in a Scientific review with “Basically” is not sounding.

Reviewer 2 Report

The Review manuscript by Rossi et al. provides an excellent overview, not only on macrophage heme oxygenase-1 (HO-1) in the context of acute kidney injury (AKI) but also on the surrounding microenvironment of the pathological setting.

The manuscript is perfectly sectioned, full of relevant information, and meticulously detailed in terms of references.

The figures are relevant and good schematics. Only suggestion: make Figure 2 bigger and include an explanatory figure legend as it is present in Figure 1.

This Reviewer has no further suggestions for improvement.
